# Definitions, Sources and Self-Reported Consumption of Regionally Grown Fruits and Vegetables in Two Regions of Australia

**DOI:** 10.3390/nu12041026

**Published:** 2020-04-08

**Authors:** Katherine Kent, Stephanie Godrich, Sandra Murray, Stuart Auckland, Lauren Blekkenhorst, Beth Penrose, Johnny Lo, Amanda Devine

**Affiliations:** 1Centre for Rural Health, University of Tasmania, Launceston, TAS 7250, Australia; stuart.auckland@utas.edu.au; 2School of Medical and Health Sciences, Edith Cowan University, Joondalup, WA 6050, Australia; s.godrich@ecu.edu.au (S.G.); l.blekkenhorst@ecu.edu.au (L.B.); a.devine@ecu.edu.au (A.D.); 3School of Health Sciences, University of Tasmania, Launceston, TAS 7250, Australia; Sandra.murray@utas.edu.au; 4Medical School, The University of Western Australia, Perth, WA 6050, Australia; 5Tasmanian Institute of Agriculture, University of Tasmania, Hobart, TAS 7000, Australia; beth.penrose@utas.edu.au; 6School of Science, Edith Cowan University, Joondalup, WA 6050, Australia; j.lo@ecu.edu.au

**Keywords:** fruit, vegetables, regional, rural, food preferences, food supply

## Abstract

Regional food systems are complex networks, with numerous retail sources that underpin a local economy. However, evidence is limited regarding how consumers define, identify, and source regionally grown fresh fruits and vegetables (RGFFV). A cross-sectional study was conducted in Tasmania (TAS) and South Western Australia (SWA) to compare how RGFFV are defined, identified and sourced by consumers, including self-reported consumption of selected RGFFV. Survey data were analyzed using the Chi-square test and *t*-tests. Results (TAS *n* = 120, SWA *n* = 123) identified that consumers had mixed perceptions of how RGFFV are defined, including produce sold at farmers markets, or grown within their region (TAS/SWA). RGFFV were commonly identified using product labelling (55% TAS, 69% SWA; *p* > 0.05). Respondents reported frequently shopping for RGFFV at major supermarkets, with more TAS respondents shopping weekly in comparison to SWA respondents (67% vs. 38%; *p* < 0.001). Supermarkets offered convenience and consumers enjoyed the experience of farmers’ markets, especially in TAS (42%) in comparison to SWA (21%; *p* = 0.012). The major RGFFV consumed were root vegetables and apples/pears, but consumers were frequently unsure about the produce’s provenance. Our findings indicate multiple opportunities to improve consumption of fresh, regional produce in TAS and SWA, which may positively impact regional economic growth and community health.

## 1. Introduction

It has been well documented that eating adequate fruit and vegetables daily may be protective against chronic diseases, including cardiovascular diseases [1] and some cancers [2]. The Australian Dietary Guidelines [3] recommend a minimum number of fruit and vegetable servings each day to ensure good nutrition and health. Despite this evidence, in 2017–2018 only 5.4% of Australian adults met both guidelines for fruits and vegetables [4]. Increasing daily fruit and vegetable consumption to 600 g could reduce the total worldwide burden of disease by 1.8% [5].

Low levels of fruit and vegetable consumption are driven by many interconnected factors [6]. The evidence for a relationship between environmental conditions and fruit and vegetable consumption is increasing [7]. Local food environments have been shown to either positively or negatively impact dietary behaviors [8]. There has been a progressive shift towards globalized food systems, which has reduced the price of foods, and minimized the impact of seasonality associated with fruits and vegetables. However, there is growing concern regarding the environmental, social, economic and food quality outcomes of globalized food systems. There is also increasing discussion centered on the impact of disasters and climate change on food security and food system sustainability. “Alternative” food systems are more localized collaborative networks integrating sustainable food production, processing, distribution, consumption and waste management to maximize the environmental, economic and social health of a region [9]. Across the world, there is growing interest in restoring strong connections between agriculture, food, environment and health, in order to support these strong local food systems and increase the health of populations [10]. 

A principle of alternative food systems is minimizing the distance between food production and consumption [11]. However, there remains no commonly agreed definition of “locally grown” or “regionally grown” food, and the definitions in the Australian context, have been under-researched. Varied definitions that are preferred by consumers have been reported [12,13,14,15,16]; some consumers may consider regionally grown food to be produced near where they live, whereas others may define it as grown in the same country where it is consumed. Despite these different definitions, research has shown that consumers have consistent expectations of regionally grown food. These include freshness, safety, high quality, and economic benefits to their community [17]. However, arguably, the lack of consensus on appropriate definitions of these foods may be preventing the growth of alternative food systems through insufficient response to evolving consumer desires [18].

Evidently, consumers prefer to use food labels to identify product attributes [19]. Consumers’ ability to identify regionally grown food has been recognized as the greatest opportunity for increasing such purchases [20]. However, it is still unclear how, and to what extent, consumers identify fruits and vegetables as regionally grown. Numerous retail outlets underpin strong alternative food systems, where local foods are not only marketed through farmers’ markets and other community-supported direct-market outlets, but also in large and small grocers and supermarkets [18]. At each type of retail outlet, exploiting the competitive advantages of regionally grown food has been promoted [21], as consumer demand for regionally grown food increases. Whilst supermarket shopping predominates in the purchasing of local food, which is linked to convenience, direct points of sale (e.g., farmers’ markets) remain important for understanding the personal connections that exist between the producer and consumer (e.g., for enjoyment), which can be far more significant in motivating behavior than the mere acquisition of products [21].

There is a significant body of research on the perceived benefits to consumers who purchase and consume regionally grown foods both internationally [22,23,24] and in Australia [25]. However, there is limited research detailing more wide-spread consumption patterns of regionally grown foods. While some research has been conducted, inconsistent methodologies have been applied, and few studies have managed to accurately estimate the amount of local food consumed. Some studies have attempted to quantify local food consumption on a population level, by evaluating census data from small-scale food businesses and data measuring food sales direct to consumers [26]. More consumer-focused research has surveyed consumption of local foods, categorizing consumers into purchasers and non-purchasers [27]. However, these data are limited and highlight a missed opportunity to quantify local food consumption using traditional nutritional assessment methodologies. Habitual food consumption is frequently determined using semi-quantitative food frequency questionnaires (SFFQ), where consumption of specific foods (with pre-defined portions) are estimated over a specified period. An adapted version of such a tool may be useful for further quantifying consumption of regionally grown foods in consumers, and to identify the specific regionally grown foods that consumers are eating. 

The importance of consumer perspective in alternative food systems research should not be underestimated, as the information associated with agricultural and business practices is limited. However, a continuing lack of consumer perspectives in alternative food systems research has been reported [28], and Australian evidence is especially limited regarding consumer perceptions of how to define, identify, and source regionally grown fresh fruits and vegetables (RGFFV) [14]. Therefore, a cross-sectional study was conducted in Tasmania (TAS) and South Western Australia (SWA), which aimed to determine and compare: (i) how RGFFV are defined and identified by consumers; (ii) where RGFFV are sourced and purchasing patterns; and (iii) self-reported consumption of selected RGFFV. 

## 2. Materials and Methods 

### 2.1. Study Sites and Participants

The study was conducted in two Australian regions; the state of TAS and the SWA region, which encompassed the South West and Great Southern regions of WA [29] (Figure 1). Despite being geographically far apart, TAS and SWA have similar fruit and vegetable production when compared to other states and territories (previously described in [25]). 

### 2.2. Questionnaire Development

A recently published manuscript [25] utilized data collected concurrently with this study and described consumer perceptions of the perceived importance of RGFFV and the barriers and enablers related to their access and consumption. 

A survey was developed for the purpose of this study to investigate RGFFV in SWA and TAS, including specific questions related to defining and identifying RGFFV (*n* = 2); how and why consumers sourced RGFFV (*n* = 13); self-reported consumption of RGFFV using a semi-quantitative food frequency questionnaire (SFFQ) developed for the purpose of this study (SWWA *n* = 23 and TAS *n* = 26); and sociodemographic information (*n* = 7). 

There are inconsistent definitions of “local food” and “regional food” in the literature [30], and it has been reported that individuals hold unique views regarding how best to define this concept. Therefore, our participants were asked to select what they felt best described RGFFV from a list of pre-defined options. These included fruits and vegetables available at supermarkets, farmers market or roadside stalls, or grown and sold within predefined regions. 

Participants were asked to select how they tried to identify RGFFV from a list of pre-defined options. These options included: (1) using food labels, (2) enquiring through a sales assistant, (3) not interested/don’t know, and (4) “other” (open-ended response option).

A seven-part question was used to determine how often participants shopped for RGFFV at seven different locations: (1) major supermarkets, (2) minor supermarkets, (3) general/corner stores, (4) fruit and vegetable shops, (5) farmers’ markets, (6) grow their own, or (7) “other” with an open-ended response. Participants were asked to tick one of six frequency options: (1) daily, (2) 2–3 times per week, (3) once per week, (4) once per fortnight, (5) once per month, (6) rarely or never.

A second seven-part question asked participants to indicate their main reason for shopping at each of the aforementioned locations, with the response options including: (1) close to where I live, (2) close to where I work, (3) it offers many choices, (4) good value for money, (5) close to public transport, (6) enjoy the experience, or (7) not applicable.

A SFFQ was developed to determine frequency of consumption of specific RGFFV, and what proportion of that food was regionally grown. In TAS, major sources of RGFFV were determined using a report of seasonal food available in TAS [31] and Eat Well Tasmania’s public “what’s in season” guide [32], resulting in the development of 19 questions related to vegetables and seven fruit-related questions. In SWA, major sources of RGFFV were determined using information available from the South-West Development Commission [33] and Buy West Eat Best South-West [ref], resulting in 14 questions related to vegetables and 11 fruit-related questions. Some nutritionally similar foods were grouped to reduce the length of the SFFQ. For example, stone fruits were one-line item, but respondents were asked to include information about apricots, nectarines, plums and peaches. Standard portion sizes were used according to the Australian Guide to Healthy Eating [34]. For example, one serving equated to half a cup for cooked leafy or dense vegetables, a full cup for raw, leafy vegetables, and/or one medium or two small pieces of fruit. Some food items had different portion sizes based on how they are typically consumed: Lemons, limes (¼ medium or 1 tbs juice/30 g), herbs/chilli/garlic (1 tbs/30 g), berries (75 g), quince (75 g), and prunes (75 g). For each fruit and vegetable, participants were asked to tick the most relevant check box for how frequently they ate that food in total (regardless of where it was grown or purchased) with six response options: (1) rarely or never, (2) 1–3 times a month or less, (3) once a week, (4) 2–4 times a week, (5) once a day, (6) 2–3 times a day. Secondly, respondents were asked to select what proportion (approximately) of that food was regionally grown, with six response options given: (1) 0%, (2) 25%, (3) 50%, (4) 75%, (5) 100%, or (6) unsure. The sociodemographic characteristics included postcode, suburb, age, gender, education, occupation, household income, household number of adults, and household number of dependents < 18 years old/children. 

To pilot test and assess the face validity of the survey tool, various stakeholders were invited to provide feedback, including academics from public health nutrition and agriculture (*n* = 2), a representative from an organization in the food and health sector (*n* = 1), and members of the general community (*n* = 5). In addition to whether the developed tool was subjectively viewed as covering the concepts it was developed to measure, stakeholders also provided feedback on survey length, question structure and formatting. The pilot-tested survey tool was amended based on their feedback prior to use in the study.

### 2.3. Data Collection

Between May and December 2018, a cross-sectional study was conducted in TAS and SWA. Adult residents (aged 18 years and over) of SWA and TAS were invited to participate by completing either a paper-based version or online version of the survey. Participants were predominantly recruited using convenience sampling through a variety of consenting community locations, including agricultural fairs, markets and libraries, where paper-based copies of surveys and flyers advertising the study were disseminated. Participants either completed the paper-based surveys on the spot and returned them to the project team or returned them at a later date by posting them in a stamped, self-addressed envelope. To complete the online version of the survey, potential participants were given a flyer that included a link to the online platform where they could complete the questionnaire online. The online recruitment strategies included disseminating the study flyer on social media sites specific to the regions; and posts on institutional media sites and e-newsletters to University staff and students. Interviews with traditional media outlets were also used to promote the survey and research. 

All participants were provided with a participant information sheet with the survey, and informed consent was implied through the return of a completed hard copy of the survey in person or the reply-paid posting of their completed survey. All participants that completed the survey online were provided with a participant information sheet at the beginning of the survey. Participants were asked to download the participant information sheet and were then asked “Have you read the information provided in the Participant Information Sheet and do you freely agree to participant in this project?”. Participants who selected “no” could not proceed. The study was conducted in accordance with the Declaration of Helsinki, and the protocol was approved by the University of Tasmania’s Tasmania Social Sciences Human Research Ethics Committee (Reference: H0017287) with multicenter approval provided by Edith Cowan University’s Human Research Ethics Committee.

The survey platforms REDCap (TAS) and Qualtrics (SWA) were used as the licensed survey platforms at each respective project team’s university. Two members of the research team (one member in TAS and SWA) entered the returned hard copy surveys into REDCap or Qualtrics. Data sets were exported from the online survey platforms to IBM SPSS Statistics for Windows, version 24.0 (IBM Corp. Armonk, NY, USA), and were screened by a third member of the team to ensure completeness. Data were then cleaned and prepared for statistical analysis. All available survey data available were used in the analyses. 

### 2.4. Data Analysis

Categorical and ordinal socio-demographic variables were cross-tabulated and summarized with frequencies and proportions. The following changes to socio-demographic variables were made due to low numbers of respondents in some categories: (1) age group was collapsed into five categories (18–30 years, 31–40 years, 41–50 years, 51–60 years and 61–75+ years), as the two oldest age groups (61–75 years and 75+ years) were combined. Income brackets were collapsed into four categories (AUD$20,000–40,000, $40,000–60,000, $60,000–80,000 and $80,000–100,000+) by combining the top two income brackets. For each household, the number of adults was collapsed into three categories (1, 2, 3+), and the number of dependents was reduced to four categories (0, 1, 2, 3+). Shopping frequencies were reduced from six categories (daily, 2–3 times per week, once per week, once per fortnight, once per month, rarely or never) to four (daily, weekly, monthly, rarely). The amount (g) of fruit or vegetable consumed per week was calculated by multiplying the portion size of each specific fruit or vegetable by the following conversion factor: rarely/never = 0, 1–3/month = 0.5, 1/week = 1, 2–4/week = 3, 1/day = 7, 2–3/day = 17.5. “Rarely/never” and “1–3/month” were collapsed into “monthly or less”, and “1/day” and “2–3/days” were collapsed into “once a day or more”. The original consumption frequency options in the SFFQ were then reduced from six frequencies to four, by collapsing the first two and first last frequency options (monthly or less, once a week, 2–3 times per week, once a day or more). 

Socio-demographic data and SFFQ variables were normally distributed. The chi-square test assessed differences in proportions for the socio-demographic variables between TAS and SWA. The chi-square test assessed differences in the proportion of respondents agreeing with different definitions of RGFFV, methods of identifying RGFFV, shopping frequencies, shopping motivators, and RGFFV consumption frequencies between TAS and SWA. An independent samples *t*-test was used to determine differences in mean intakes of each fruit and vegetable (g per week) between TAS and SWA. The significance level for all analyses was set at *p* ≤ 0.05.

## 3. Results

### 3.1. Socio-Demographics 

Survey data from respondents in TAS (*n* = 120) and SWA (*n* = 123) were collated and analyzed. No significant differences were observed between TAS and SWA for most socio-demographic variables including age, gender, education and household income (*p* > 0.05) (Table 1). The households of the SWA respondents had fewer adults than the TAS respondents (*p* = 0.018). Only 75% of TAS respondents were the main household shoppers in comparison to SWA respondents (94%) (*p* < 0.001).

### 3.2. How RGFFV are Defined and Identified in TAS and SWA

TAS respondents were more likely to check multiple responses to this question in comparison to SWA respondents, corresponding with significantly higher percentage agreement (all *p* < 0.05) across all the definitions in TAS (Figure 2). The percentage agreement was highest for the definition of fruits and vegetables available at local farmers’ markets (53% TAS and 21% SWA), followed by the definition of RGFFV produced within the region (19% SWA and 35% TAS). Fewer respondents agreed with definitions of RGFFV as being foods produced within a 50 km radius of their area (10% in SWA and 28% in TAS) or fruits and vegetables available through roadside stalls (1% in SWA and 19% in TAS). 

Respondents were most likely to report using food labels to identify RGFFV (Table 2), in both TAS (55%) and SWA (69%). A chi-square test indicated there were significant differences in TAS and SWA respondents’ methods used to identify RGFFV (*p* < 0.001), with TAS respondents more likely to be unsure of how to identify these foods (Table 2). Respondents who checked “other” and provided an open-ended response either referred to growing their own produce (and therefore knew it was grown in the region) or shopping specifically at a location that only stocked regionally grown food (e.g., a regional fruit and vegetable shop).

A chi-square test indicated there was no significant difference in the respondents’ definitions of RGFFV according to their sociodemographic characteristics, including age, education level, number of adults or dependents in the household, or income (all *p* > 0.05).

### 3.3. Where and Why RGFFV are Sourced in TAS and SWA

The shopping frequency of respondents in TAS and SWA at each of the shop locations is displayed in Figure 3. Results indicated that TAS respondents shopped more frequently (weekly: 67%) at major supermarkets in comparison to SWA respondents (weekly: 38%), (*p* < 0.001). More TAS respondents reported shopping monthly at fruit and vegetable shops in comparison to more SWA respondents who reported shopping rarely (*p* = 0.012). There was no significant difference in shopping frequencies at minor supermarkets, general/corner stores, farmers’ markets, farm gate sales or for those who accessed the fruits and vegetables they grew themselves (all *p* = 0.05). 

A chi-square test investigated whether shopping frequencies reported by the respondents differed according to their sociodemographic characteristics, including age, education level, and number of adults or dependents in the household. Significant differences were identified in the shopping frequency at general/corner stores by age, where younger respondents shopped more frequently than older adults (*p* = 0.034). Older adults reported more frequently accessing produce they grew on their own (*p* = 0.001) in comparison to younger respondents. Respondents with secondary level education shopped more frequently at general/corner stores than those with tertiary education (*p* = 0.003). Lastly, those households with three or more dependents shopped more frequently at major supermarkets in comparison to those households with fewer (or no) dependents, who predominantly shopped weekly (*p* = 0.004). There were no other significant differences in shopping frequencies according to sociodemographic characteristics (all *p* > 0.05). The major motivating reason for shopping at each shop location is displayed in Figure 4. Being close to public transport was not a major motivating factor for shopping at any location, with this reason contributing 1% or less for any shopping outlet. Respondents in both TAS and SWA were most likely to report that a shop being close to where they live as the main motivating factor for shopping there, especially for major supermarkets with 54% and 35% reporting this in TAS and SWA, respectively. In TAS and SWA, respondents reported enjoying the experience of both farmers’ markets and growing their own fruits and vegetables. However, for major supermarkets, more TAS respondents (55%) reported proximity to where they live as the main motivating factor for shopping in major supermarkets as compared to SWA respondents (35%) (*p* < 0.001). Additionally, more TAS respondents reported enjoying the experience of farmers’ markets (42%) in comparison to SWA respondents (21%) (*p* = 0.012). There was no significant difference in major motivating factors for shopping at minor supermarkets, general/corner stores, fruit and vegetable shops, farm gates or for those who grew their own (all *p* ≥ 0.05). 

A chi-square test investigated whether motivating factors for shopping reported by the respondents differed according to their sociodemographic characteristics (age, education level, and number of adults or dependents in the household). Significant differences in the major motivating factors included that respondents with higher incomes reported enjoying the experience of farmers’ markets, and those with lower incomes reported they were good value for money (*p* = 0.027). Older respondents reported choosing to “grow their own” as the produce was close to where they lived, whereas a higher proportion of younger respondents reported that they enjoyed the experience (*p* = 0.018). Lastly, respondents with no dependents were more likely to shop for RGFFV at general/corner stores because they were close to where they lived, whereas those with three or more dependents reported enjoying the experience (*p* = 0.039). There were no other significant differences in shopping frequencies according to sociodemographic characteristics (all *p* > 0.05).

### 3.4. Self-Reported Consumption of Major RGFFV in TAS and SWA

Frequency of consumption of major RGFFV grown in TAS and SWA are reported in Appendix A. The most commonly consumed fruit item was apples and pears with 47% of TAS respondents and 32% of SWA respondents consuming them daily. Herbs and root vegetables were the most commonly consumed vegetables for TAS respondents, with 50% and 40% of respondents consuming them daily. In SWA, respondents reported most commonly consuming brassica vegetables and leafy greens, with 28% and 29% of respondents reporting consuming them daily. Significant differences in frequencies of consumption were observed between TAS and SWA for beans/peas (*p* < 0.001), root vegetables (*p* = 0.008), corn (*p* = 0.001), leeks/onions/shallots (*p* = 0.002), leafy greens (*p* = 0.003), potatoes (*p* = 0.021) and herbs (*p* < 0.001), where TAS respondents reported consuming them more frequently than SWA respondents.

The frequency of consumption was converted to intake in g per week (using the portion size) for major RGFFV consumed in TAS and SWA (reported in Table 3). Of all vegetables, TAS respondents consumed the most root vegetables, consuming nearly 350 g per week. This was significantly different (*p* = 0.002) to SWA, who only consumed around 240 g of these vegetables per week. SWA respondents consumed the most broccoli (and other brassica vegetables), consuming around 290 g per week, which was similar for TAS respondents (270 g per week). TAS respondents reported consuming significantly more beans/peas, leeks/onions, and herbs than SWA respondents (Table 3). Of all fruit items, both SWA and TAS respondents reported consuming the greatest number of apples/pears (Table 3). However, this was significantly different between sites with TAS respondents consuming 772 g per week and SWA respondents consuming 590 g per week. SWA respondents consumed significantly more lemons (*p* = 0.012), but consumption of other fruit was similar between the sites. 

Table 4 identifies the proportion of these fruits and vegetables estimated to be regionally grown and shows a wide range of proportions reported across all the fruits and vegetables. Of the vegetables, TAS respondents reported being unsure about the origin of celery (56% of the time), radishes (54% of the time), and capsicums (48% of the time). SWA respondents were more certain about the origin of their food overall, but reported being unsure about corn, celery and asparagus/artichokes (all 34% of the time). Of the fruit, TAS respondents reported being unsure about the origin of passionfruit and figs 50% and 55% of the time, respectively. In SWA, respondents reported being unsure about the origin of persimmons and prunes, at 57% and 43% of the time, respectively. Both, TAS and SWA respondents were least unsure about the origin of apples and pears, with respondents only unsure around 20% of the time (Table 4). Furthermore, significant differences in the proportions of fruit and vegetables consumed by respondents in TAS and SWA estimated as being regionally grown were observed for carrots (*p* = 0.027), capsicum (*p* = 0.001), celery (*p* = 0.006), stone fruits (*p* = 0.016), berries (*p* = 0.048). SWA respondents were more likely to consume 100% SWA grown carrots, capsicum, celery and stone fruits in comparison to TAS respondents. The opposite was found for berries, where TAS respondents were more likely to consume 100% TAS grown in comparison to SWA respondents. 

## 4. Discussion

This cross-sectional survey aimed to understand how consumers in two Australian agriculturally productive regions defined and identified RGFFV, where they sourced these products and why, and their self-reported consumption in the context of their diet. In both TAS and SWA, consumers were most likely to define RGFFV as fruits and vegetables available at farmers’ markets or produced within their specific region. These findings are somewhat at odds with international literature, where most often consumers define “local food” in terms of a pre-defined distance from their home (e.g., within 100 miles) [35]. However, Australian farmers’ markets may have stricter guidelines than in many other countries about what can be sold at a farmers’ market due to Australia’s stringent food safety standards applied at the local and state government level [36]. Produce must be sold by “the farmer/producer, family member, or employee directly involved with the growing, rearing, catching or manufacturing of the product” and “resellers are not permitted at the Farmers’ Market and the reselling of produce is not permitted at the Farmers’ Market”, which may explain the difference in our findings and previous studies [37]. While farmers’ markets in TAS and SWA generally promote the sale of regionally grown produce, and most produce is sold by the producer, it is possible that re-selling approaches exist: for example, stallholders purchasing products including produce grown outside the regional area from either producers or wholesale markets and re-selling them at farmers’ markets. This has been identified as an area of concern for stallholders at farmers’ markets in Australia [38], because of the potential for a consumer to purchase produce from a stallholder, and either be unaware that they didn’t buy from a farmer, or find out after-the-fact, potentially harming the reputation of farmers’ markets. 

Given the various decisions that a consumer makes when purchasing food, a clear definition of what constitutes “local food” is beneficial, and the lack of consensus in both TAS and SWA respondents highlights opportunities for growers and producers with these regions to clearly define and promote one united definition informed by consumer perspectives. Maintaining the status quo, by letting consumers decide for themselves how to define these foods may contribute to confusion and potentially harm the value, worth and reputation of RGFFV in TAS and SWA. Those selling RGFFV must remain aware of this lack of consensus amongst consumers and to simply label an item as “local” may be an insufficient solution. Eden, Bear, and Walker (2008) [39] have suggested better consumer-focused strategies would involve more deeply rooted educational and resource-intensive initiatives aimed at allowing consumers to reclaim the capacity to know what constitutes good food, and the benefits of buying regionally grown produce. They suggest that programs that support and enable individuals to grow their own fruits and vegetables may be a mechanism for consumers to be less dependent on food shopping at retail outlets and instill an appreciation for the quality of RGFFV. However, such approaches are complicated and would involve rethinking a whole host of social institutions [39], which may be too resource intensive for consideration with the regions of TAS and SWA specifically.

Not only understanding what local food is but being able to identify these foods requires consumers to have a sound understanding of both seasonality and what is grown in their region. In our study, most consumers identified RGFFV through product labels in both TAS and SWA. It has been reported that consumers who often looked at labels to see where a product was grown were more likely to seek local agricultural produce and pay a higher price [40]. In line with research that suggests that the vast majority of consumers want to know where their produce comes from, in Australia it is mandated that all staple foods have “country of origin” labels, which vary according to whether the food was (a) grown, produced or made in Australia, (b) packed in Australia, or (c) imported into Australia. However, more localized provenance labelling of fruits and vegetables is not commonplace in TAS or SWA in the same way specialty products such as wines and cheeses are. This gap, alongside our findings, shows opportunities for producers in these regions to campaign for clearer and more prominent provenance labelling.

Respondents reported shopping for RGFFV at larger retail stores, including major and minor supermarkets. However, there were differences in the shopping habits of respondents in TAS and SWA, where TAS consumers were likely to shop more frequently at major supermarkets than those in SWA. This finding is inconsistent with our consumer perceptions of how RGFFV are defined, given that respondents were most likely to identify that RGFFV were best defined as available at farmers’ markets. This finding also highlights the growing influence that of major retailers can have in supporting local food systems [18] as the consumer demand for regionally grown produce increases. In the USA, large retail stores have reported sourcing locally grown produce in line with their respective national marketing initiatives [18]. While it has been reported that larger food retailers have created tension with their involvement in local food systems, there is an opportunity for these types of outlets to contribute to the aggregation of local produce and may support local food systems due their economies of scale. However, their involvement in alternative food systems must be managed appropriately, as it has been argued that, currently, large retailers are causing negative economic, environmental and social effects resulting in the marginalization, inequality and vulnerability of small family farms [41]. The reliance on these outlets by our study respondents also indicates what a major motivating factor convenience is when making food choices. In our study, both TAS and SWA respondents reported accessing these retail outlets, since they were close to where they live and offered many choices, but no respondents reported enjoying the experience of shopping at these locations. Conversely, consumers in our study were most likely to enjoy the experience of farmers’ markets and growing their own fruits and vegetables. This aligns with published literature [42] that suggests farmers’ markets are social places and offer a different type of shopping experience than supermarkets or other retail outlets. For consumers, food quality remains the most important motivator for consumer purchases. Other positive attributes related to buying regionally grown produce at farmers’ markets also include to financially support local farmers and their community [25,42]. 

In our study, respondents with higher incomes reported enjoying the experience of farmers’ markets, and those with lower incomes reported that they were good value for money. This finding is in contrast with published literature, in which farmers’ markets have been criticized for passively excluding disadvantaged groups [43], and that promoting these outlets as the major retail sources for RGFFV may exclude population groups who arguably have the poorest diets and may benefit the most from accessing and consuming more RGFFV. While the cost of RGFFV at farmers’ markets is not known, this finding is a positive indication that consumers in in TAS and SWA may be able to access to affordable RGFFV at farmers’ markets in these regions. Widespread promotion of where RGFFV are available in each region, highlighting the opportunities for purchasing these foods through multiple retail outlets, may support their consumption and help consumers identify and source these foods.

A number of sociodemographic characteristics influenced consumer shopping behaviors. Notably, age influenced the reported frequency of respondents growing their own produce, with older adults more likely to grow their own than younger respondents. The benefits of gardening for older adults has been reviewed [44], showing that gardening is associated with increased overall health and quality of life. Our study builds upon these findings by examining the reasons why older adults grow their own produce. While all respondents reported enjoying the experience, older adults reported valuing the easy access to these foods (being close to where they live) more frequently than younger respondents. This finding is interesting and aligns with research that indicates that there are differences in the motivations for shopping behaviors between younger and older adults [45]. 

Examination of the self-reported consumption of selected RGFFV in TAS and SWA shows a high consumption of root vegetables, leafy greens, onions and potatoes (Table 3). Of interest is the proportion of these vegetables that were estimated to be regionally grown (Table 4); a high proportion of these food items were estimated to be 100% regionally grown, and consumers were less likely to report being unsure where these food items were grown. Similarly, for fruit, apples and pears were most commonly consumed, followed by citrus fruits (in SWA), and respondents were likely to report that 100% of these foods were regionally grown. For fruits and vegetables consumed less frequently and therefore contributing a lower amount to overall intake (as reported in Table 3), consumers were more likely to report being unsure where these foods were grown (e.g., passionfruit and prunes). Consumers have reported wanting to know more information about their food [19], and the proportion of our respondents who were unsure about the origins of selected food items provides opportunities for the producers of these foods to ensure there is clear communication around their provenance. 

### 4.1. Opportunities

The results of our research indicate that, while there are some differences in consumer shopping behaviors between TAS and SWA, there are multiple opportunities to harness change that are consistent in both TAS and SWA to improve the access and consumption of RGFFV:Clearly describing and promoting a standard definition of what RGFFV are in TAS and SWA;Where possible, using consistent product labelling and signage to clearly identify and promote RGFFV in TAS and SWA;Widespread promotion of the various outlets where RGFFV are sold in TAS and SWA to show they are sold in outlets beyond farmers’ markets;Supporting retail outlets with appropriate product promotion of regionally grown foods and encouraging clear provenance labelling in stores, especially for those RGFFV where respondents reported being unsure of the food’s provenance.

### 4.2. Strengths and Limitations

The strengths of this study included the investigation of consumer perceptions about RGFFV in two demographically similar regions (TAS and SWA) in rural Australia, which allows comparison between regions. While there were some discrepancies between shopping behaviors and consumption patterns that may be regionally specific, our study predominantly shows similarities between the two study sites. This may be related to similar levels of agricultural production between the two regions, or similarities in the sociodemographic characteristics of survey respondents. To the authors’ knowledge, this is the first study that attempts to quantify consumption of specific RGFFV using traditional nutritional assessment techniques. SFFQs are well-validated tools that can be used to determine quantities of foods consumed, which can provide specific feedback for primary producers about how much of their product consumers believe is sourced regionally. The study limitations include the use of convenience sampling and a non-validated survey tool; despite this, the tool was assessed for face and content validity. The use of a convenience sample may mean that the survey results may not be generalizable to other Australian regions. Further, the sampling approach using both face-to-face and online methods may have influenced the type of participants who consented to participate and potentially biased the results. For example, participants recruited at agricultural fairs may consume a greater amount of regionally grown produce in comparison to other individuals. Due to the sampling and data entry methods utilized in this study, a comparison of respondent characteristics (e.g., demographics or RGFFV consumption patterns) could not be performed, but the mixed sampling methods could have influenced the results. The survey was collected between May and December, which may have influenced the reported consumption data, as regionally grown food in both TAS and SWA is highly seasonal. Data was not collected in the warmer, summer months, where a large amount of fruit is grown and sold, and it is therefore possible that these foods may be underreported, and winter crops (e.g., leafy greens) may be over-represented. To account for this, participants were asked to estimate their average intake over the past 12 months; however, studies of the impact of seasonality on fruit and vegetable assessment using a SFFQ have identified significant seasonal changes in dietary intake. Therefore, these results should be interpreted in light of these limitations [46]. 

## 5. Conclusions

Our study contributes Australian findings from two agriculturally productive regions to the international literature regarding consumer perceptions of how to define, identify and source RGFFV. Our findings highlight that there are varied opinions on how consumers define RGFFV, but most agree that they are available at local farmers’ markets and they are identified through clear provenance labels. Consumers reported purchasing RGFFV at numerous retail sources with supermarkets offering convenience, but consumers enjoying direct-market sales and growing their own produce. Selected fruits and vegetables were largely identified as regionally grown, but, often, consumers were unsure about the origin of their food. These findings assist in identifying gaps and opportunities for improving the consumption of fresh produce in TAS and SWA, which may positively influence regional economic growth and community health and wellbeing.

## Figures and Tables

**Figure 1 nutrients-12-01026-f001:**
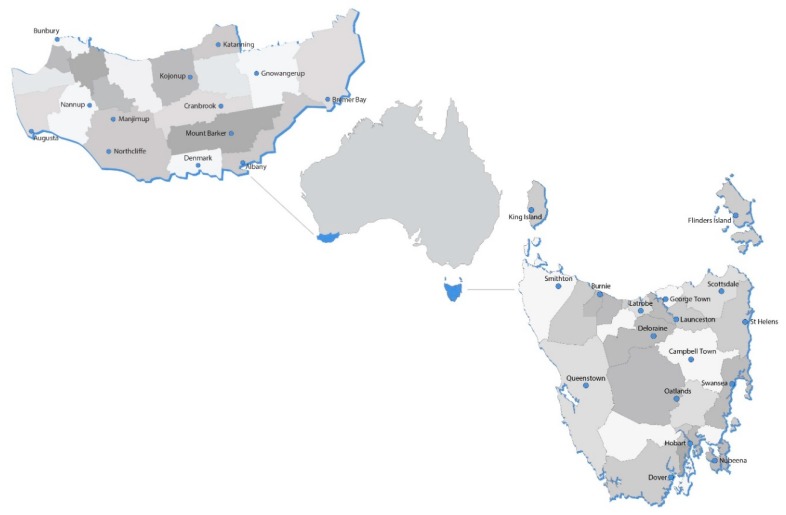
Geographical location of Tasmania and South Western Australia regions of Australia.

**Figure 2 nutrients-12-01026-f002:**
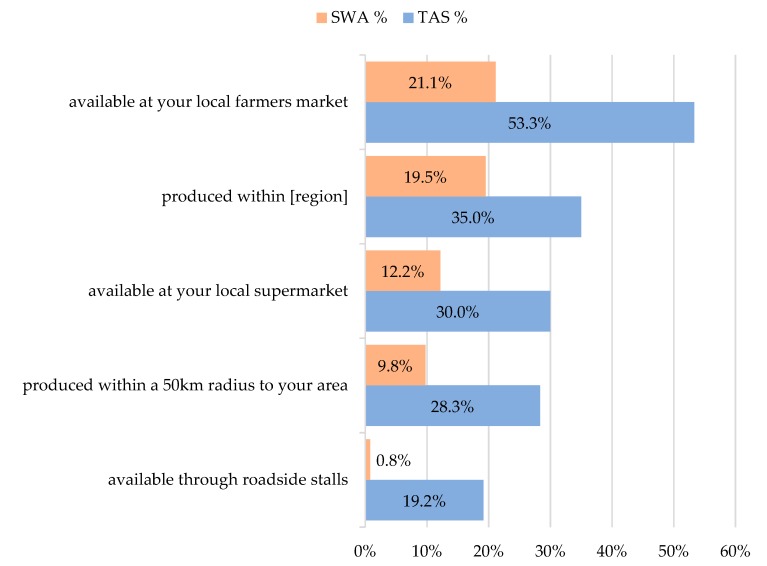
The percentage of respondents who agreed with each definition of regionally grown fresh fruits and vegetables in Tasmania (TAS) and South Western Australia (SWA) (Respondents could select more than one response).

**Figure 3 nutrients-12-01026-f003:**
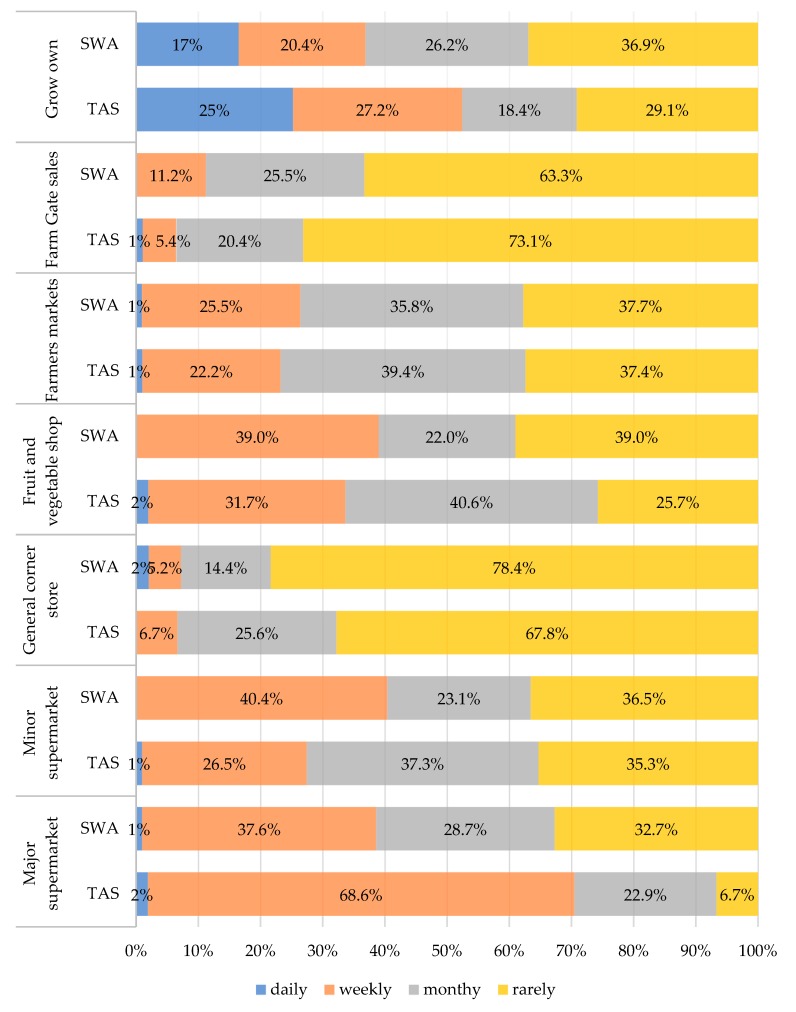
Frequency of shopping for regionally grown fresh fruits and vegetables from various food outlets in (a) Tasmania (TAS) and (b) South Western Australia (SWA).

**Figure 4 nutrients-12-01026-f004:**
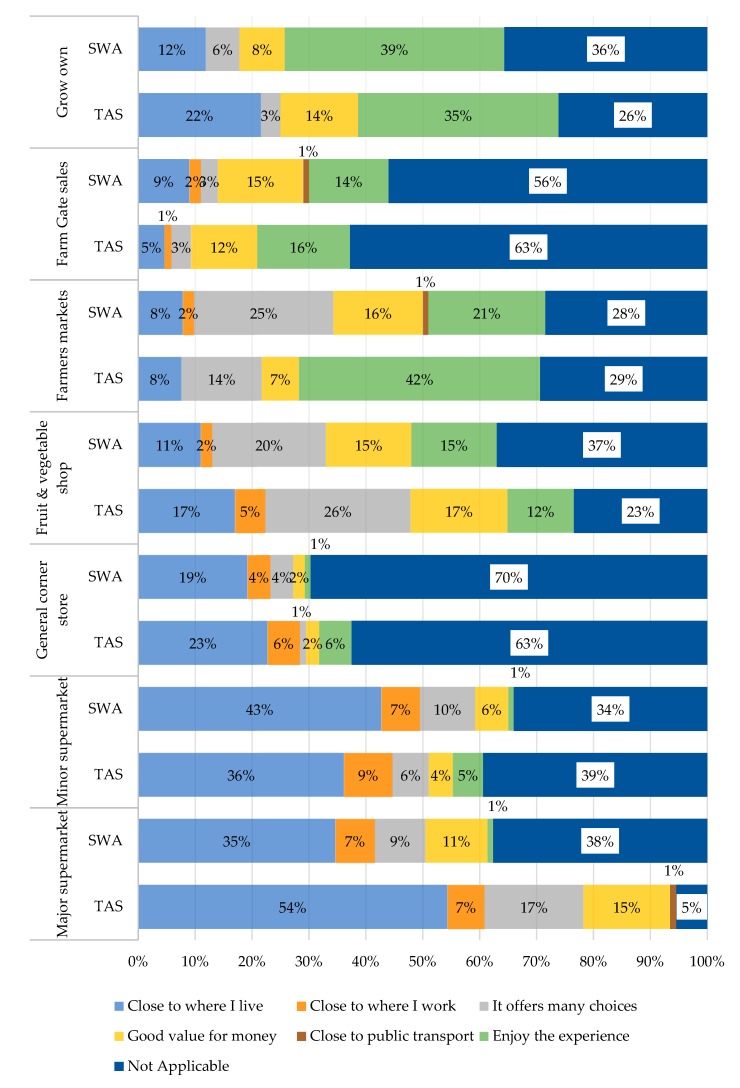
Motivating reasons for purchasing regionally grown foods from each shopping outlet in (a) Tasmania (TAS) and (b) South Western Australia (SWA).

**Table 1 nutrients-12-01026-t001:** Participant demographic characteristics in Tasmania and SWA.

		Tasmania *n* (%)	SWA *n* (%)	Total	*p*-Value
Age	18–30	27 (23.1)	14 (15.6)	41 (19.8)	0.642
	31–40	20 (17.1)	17 (18.9)	37 (17.9)	
	41–50	25 (21.4)	17 (18.9)	42 (20.3)	
	51–60	16 (13.7)	15 (16.7)	31 (15.0)	
	61+	29 (24.8)	27(30.0)	56 (27.1)	
Sex	Male	36 (30.5)	20 (22.2)	56 (26.9)	0.182
	Female	82 (69.5)	70 (77.8)	152 (73.1)	
Education	Secondary	21 (17.8)	24 (26.7)	45 (21.6)	0.124
	Tertiary	97 (82.2)	66 (73.3)	163 (78.4)	
Income	<20,000–40-000	16 (14.3)	18 (20.5)	34 (17.0)	0.507
	40,000–60,000	16 (14.3)	16 (18.2)	32 (16.0)	
	60,000–80,000	20 (17.9)	14 (15.9)	34 (17.0)	
	80,000–100,000+	60 (53.6)	40 (45.5)	100 (50.0)	
Adults in household	1	10 (8.5)	19 (21.1)	29 (13.9)	0.018
	2	82 (69.5)	59 (65.6)	141 (67.8)	
	3 or more	26 (22.0)	12 (13.3)	38 (18.3)	
Dependents in household	0	60 (54.1)	58 (65.9)	118 (59.3)	0.244
	1	14 (12.6)	12 (13.6)	26 (13.1)	
	2	26 (23.4)	13 (14.8)	39 (19.6)	
	3 or more	11 (9.9)	5 (5.7)	16 (8.0)	
Main shopper	Yes	87 (75.0)	104 (93.7)	191 (84.1)	<0.001
	No	29 (25.0)	7 (6.3)	36 (15.9)	

*p*-value derived from chi-square statistic.

**Table 2 nutrients-12-01026-t002:** Methods used to identify RGFFV in Tasmania and SWA.

	TAS *n* (%)	SWA *n* (%)
Using food labels	66 (55)	79 (69)
Enquiring through a sales assistant	6 (5)	20 (18)
I don’t know how to identify which foods are regionally grown	30 (25)	3 (3)
Other	18 (15)	12(11)

**Table 3 nutrients-12-01026-t003:** Consumption (g per week) of selected regionally grown fruit and vegetables TAS and SWA.

	TAS			SWA			
	*n*	Mean	sd	*n*	Mean	sd	*p*-Value
Asparagus, artichoke	115	51.8	79.3	92	76.6	179.0	0.185
Asian greens	109	52.6	76.7				-
Bean, peas	116	216.3	191.8	93	120.2	179.4	<0.001
Beetroot	115	75.3	101.5				-
Broccoli, brussels sprouts, cauliflower, cabbage	117	269.6	170.4	94	292.4	254.1	0.437
Carrots, parsnips, turnips, swede, fennel	116	347.8	270.4	94	242.2	198.1	0.002
Capsicum	117	173.1	181.0	91	158.7	218.9	0.604
Celery	115	118.0	164.6	92	110.5	211.4	0.772
Corn	113	111.5	112.6	94	113.3	216.9	0.939
Cucumber	113	163.6	154.6				-
Leeks, onions, shallots, spring onion	113	290.7	224.2	91	206.0	235.5	0.009
Leafy greens	115	335.9	265.8	92	289.8	304.1	0.247
Mushrooms	113	136.1	125.5				-
Potatoes	113	233.6	179.2	93	223.8	233.8	0.733
Pumpkin	114	129.3	123.4	93	171.0	221.7	0.089
Radishes	109	27.9	68.7				-
Tomatoes	112	253.8	235.6	93	261.7	257.2	0.819
Zucchini, squash, eggplant	110	151.4	146.3	92	137.4	216.9	0.586
Herbs and spices	113	161.9	136.7	90	100.2	123.7	0.001
Apples, pears	112	772.8	638.6	89	590.7	580.0	0.038
Stone fruits	113	354.4	536.2	87	300.0	451.7	0.447
Lemons	112	57.7	74.7	87	92.8	119.1	0.012
Berries, cherries	113	209.4	251.5	88	210.1	243.1	0.985
Passionfruit	107	29.1	78.0				-
Figs	108	66.7	274.2				-
Quince, rhubarb	111	30.1	65.5				-
Avocado				87	216.8	274.9	-
Citrus				89	477.8	530.6	-
Grapes				87	189.7	347.4	-
Kiwi fruit				87	120.7	217.4	-
Persimmons				87	33.6	132.3	-
Prunes				88	24.7	73.0	-

Berries include blackberry, blueberry, gooseberry, raspberry, strawberry, tayberry, yosterberry; stone fruits include apricot, greengage, nectarine, peach, plum; herbs include chilli, garlic, thyme, parsley, coriander, rosemary, oregano, chives, basil, sage, mint; leafy greens include lettuce, rocket, sprouts chard/silver beet, kale, spinach, mustard greens; *p*-value from independent samples *t*-test; n is the number of respondents for each food item.

**Table 4 nutrients-12-01026-t004:** Proportion of fruit and vegetables consumed by respondents in TAS and SWA estimated as being regionally grown.

		TAS n (%)							SWA n(%)					
	Portion Size	0%	25%	50%	75%	100%	Unsure	0%	25%	50%	75%	100%	Unsure	*p*-Value
Asparagus, artichoke	1/2 cup	10	4	6	2	31	46	13	4	7	7	28	30	0.282
		(10.1)	(4.0)	(6.1)	(2.0)	(31.3)	(46.5)	(14.6)	(4.5)	(7.9)	(7.9)	(31.5)	(33.7)	
Asian greens	1/2 cup	15	4	6	8	21	46	-	-	-	-	-	-	-
		(15.0)	(4.0)	(6.0)	(8.0)	(21.0)	(46.0)							
Bean, peas	1/2 cup	3	4	25	10	31	35	8	3	11	11	30	29	0.202
		(2.8)	(3.7)	(23.1)	(9.3)	(28.7)	(32.4)	(8.7)	(3.3)	(12.0)	(12.0)	(32.6)	(31.5)	
Beetroot	1/2 cup	5	6	6	9	33	42	-	-	-	-	-	-	-
		(5.0)	(5.9)	(5.9)	(8.9)	(32.7)	(41.6)							
Broccoli, brussels sprouts, cauliflower, cabbage	1/2 cup	0	6	19	13	39	32	1	1	9	14	45	24	0.122
		(0)	(5.5)	(17.4)	(11.9)	(35.8)	(29.4)	(1.1)	(1.1)	(9.6)	(14.9)	(47.9)	(25.5)	
Carrots, parsnips, turnips, swede, fennel	1/2 cup	0	6	18	13	42	33	2	2	6	21	41	22	0.027
		(0)	(5.4)	(16.1)	(11.6)	(37.5)	(29.5)	(2.1)	(2.1)	(6.4)	(22.3)	(43.6)	(23.4)	
Capsicum	1/2 cup	10	7	12	9	19	53	3	1	11	10	38	29	0.001
		(9.1)	(6.4)	(10.9)	(8.2)	(17.3)	(48.2)	(3.3)	(1.1)	(12.0)	(10.9)	(41.3)	(31.5)	
Celery	1/2 cup	7	6	6	7	20	60	5	2	11	8	35	31	0.006
		(6.6)	(5.7)	(5.7)	(6.6)	(18.9)	(56.6)	(5.4)	(2.2)	(12.0)	(8.7)	(38.0)	(33.7)	
Corn	1/2 cup	10	10	9	8	25	48	8	5	7	12	30	32	0.372
		(9.1)	(9.1)	(8.2)	(7.3)	(22.7)	(43.6)	(8.5)	(5.3)	(7.4)	(12.8)	(31.9)	(34.0)	
Cucumber	1/2 cup	5	3	11	14	28	48	-	-	-	-	-	-	-
		(4.6)	(2.8)	(10.1)	(12.8)	(25.7)	(44.0)							
Leeks, onions, shallots, spring onion	1/2 cup	2	2	14	17	40	34	2	2	5	15	41	27	0.58
		(1.8)	(1.8)	(12.8)	(15.6)	(36.7)	(31.2)	(2.2)	(2.2)	(5.4)	(16.3)	(44.6)	(29.3)	
Leafy greens	1 cup fresh	2	5	7	16	49	31	1	1	11	11	46	24	0.47
		(1.8)	(4.5)	(6.4)	(14.5)	(44.5)	(28.2)	(1.1)	(1.1)	(11.7)	(11.7)	(48.9)	(25.5)	
Mushrooms	1/2 cup	3	4	8	7	41	44	-	-	-	-	-	-	-
		(2.8)	(3.7)	(7.5)	(6.5)	(38.3)	(41.1)							
Potatoes	1/2 cup	0	1	8	17	65	19	1	3	8	10	51	21	0.497
			(0.9)	(7.3)	(15.5)	(59.1)	(17.3)	(1.1)	(3.2)	(8.5)	(10.6)	(54.3)	(22.3)	
Pumpkin	1/2 cup	1	1	8	15	48	33	0	1	10	11	51	20	0.509
		(0.9)	(0.9)	(7.5)	(14.2)	(45.3)	(31.1)	(0)	(1.1)	(10.8)	(11.8)	(54.8)	(21.5)	
Radishes	1/2 cup	8	1	1	7	25	50	-	-	-	-	-	-	-
		(8.7)	(1.1)	(1.1)	(7.6)	(27.2)	(54.3)							
Tomatoes	1/2 cup	1	5	12	14	39	33	1	3	11	17	44	18	0.409
		(1.0)	(4.8)	(11.5)	(13.5)	(37.5)	(31.7)	(1.1)	(3.2)	(11.7)	(18.1)	(46.8)	(19.1)	
Zucchini, squash, eggplant	1/2 cup	4	3	9	17	35	37	-	-	-	-	-	-	-
		(3.8)	(2.9)	(8.6)	(16.2)	(33.3)	(35.2)							
Herbs and spices	1 tb	9	5	6	10	39	37	8	5	7	14	30	29	0.834
		(8.5)	(4.7)	(5.7)	(9.4)	(36.8)	(34.9)	(8.6)	(5.4)	(7.5)	(15.1)	(32.3)	(31.2)	
Apples, pears	1 medium	0	1	4	13	66	23	2	1	5	7	57	18	0.587
		(0)	(0.9)	(3.7)	(12.1)	(61.7)	(21.5)	(2.2)	(1.1)	(5.6)	(7.8)	(63.3)	(20.0)	
Stone fruits	1 medium or 2 small	1	6	9	10	37	39	2	3	2	11	51	19	0.016
		(1.0)	(5.9)	(8.8)	(9.8)	(36.3)	(38.2)	(2.3)	(3.4)	(2.3)	(12.5)	(58.0)	(21.6)	
Lemons	¼ lemon or 1 tb juice	4	1	4	10	52	37	3	0	1	6	60	17	0.082
		(3.7)	(0.9)	(3.7)	(9.3)	(48.1)	(34.3)	(3.4)	(0)	(1.1)	(6.9)	(69.0)	(19.5)	
Berries	1/2 cup	1	7	8	12	58	24	7	6	5	5	37	29	0.048
		(0.9)	(6.4)	(7.3)	(10.9)	(52.7)	(21.8)	(7.9)	(6.7)	(5.6)	(5.6)	(41.6)	(32.6)	
Passionfruit	1 medium or 2 small	13	2	3	0	24	52	-	-	-	-	-	-	-
		(13.8)	(2.1)	(3.2)	(0)	(25.5)	(55.3)							
Figs	1 medium or 2 small	13	3	0	2	30	48	-	-	-	-	-	-	-
		(13.5)	(3.1)	(0)	(2.1)	(31.3)	(50.0)							
Quince, rhubarb	½ cup cooked	13	0	0	1	42	43	-	-	-	-	-	-	-
		(13.1)	(0)	(0)	(1.0)	(42.4)	(43.4)							
Avocado	½ medium	-	-	-	-	-	-	3	4	4	10	45	21	-
								(4.6)	(4.6)	(4.6)	(8.0)	(46.0)	(32.2)	
Citrus	1 medium	-	-	-	-	-	-	1	3	3	6	53	24	-
								(1.1)	(3.3)	(3.3)	(6.7)	(58.9)	(26.7)	
Grapes	1 cup	-	-	-	-	-	-	5	3	6	8	37	29	-
								(5.7)	(3.4)	(6.8)	(9.1)	(42.0)	(33.0)	
Kiwi fruit	2 small	-	-	-	-	-	-	15	4	9	7	19	33	-
								(17.2)	(4.6)	(10.3)	(8.0)	(21.8)	(37.9)	
Persimmons	1 medium	-	-	-	-	-	-	17	0	2	6	23	36	-
								(20.2)	(0)	(2.4)	(7.1)	(27.4)	(42.9)	
Prunes	½ cup	-	-	-	-	-	-	21	2	3	2	8	48	-
								(25.0)	(2.4)	(3.6)	(2.4)	(9.5)	(57.1)	

Berries include blackberry, blueberry, cherry, gooseberry, raspberry, strawberry, tayberry, yosterberry; stone fruits include apricot, greengage, nectarine, peach, plum; herbs include chilli, garlic, thyme, parsley, coriander, rosemary, oregano, chives, basil, sage, mint; leafy greens include lettuce, rocket, sprouts chard/silver beet, kale, spinach, mustard greens; *p*-value derived from chi-square statistic; n is the number of respondents for each food item; portion size is derived from Australian Guide to Healthy Eating [34].

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
