# Peer review of "Definitions, Sources and Self-Reported Consumption of Regionally Grown Fruits and Vegetables in Two Regions of Australia"

_nutrients, 2020, doi:10.3390/nu12041026_

Round 1
Reviewer 1 Report
The manuscript, "Definitions, sources, and self-reported consumption of regionally grown fruits and vegetables in two regions of Australia," captures the perspectives of consumers of how to define, identify, and and sources regionally grown fresh fruits and vegetables (RGFVV). This study compares consumers in two regions Tasmania (TAS) and South West Australia (SWA).
Methods:
Line 146 "1 serve" should this be "1 serving"?
Line 167 May-December 2018, Please address the seasonal variation in availability of local produce in terms of amounts and types of fruits and vegetables. How might this affect the results?
Line 170 Sampling. Was there a difference between those who completed surveys on paper versus those who completed surveys on line, demographics such as age, sex, income difference? and in terms of responses? and how might this affect the results?
Line 178 Informed Consent. Was this different for online versus in person interviews?
Results:
Were there difference between younger and older consumers on perceptions? which has been found in other studies (see Hsiao et al. 2018 doi: 10.1017/S1368980017003755)
Figures 3 and 4: These are difficult to read and interpret, bar charts for each item or perhaps a pie chart would work better?
Table 3: what is "n"? grams per week? Please make this clear in the table.
Table 4: Supplemental Table?
Discussion:
Line 314 "at farmers markets" - whether or not the foods were grown regionally. In these areas of Australia do farmers markets sell non-regionally grown foods like many other places do?
Conclusions:
Some of the conclusions feel overstated
Line 413 "contributes valuable Australian findings to international literature" as the authors note in the limitations these findings are limited in their generalizability.
Line 415 "novel approach to quantify the consumption" the methods used seem commonplace, and yet used a non-validated tool?
Overall, the study contributes to potential marketing gaps that may be filled.
Reviewer 2 Report
Study contributes valuable Australian consumer perceptions how RGFFV are defined and identified, Authors recognize where RGFFV are sourced and identified purchasing patterns RGFFV in two rural regions of Australia and also Authors researched self-reported consumption of selected RGFFV by respondents. Nowadays the topic is relevant because of alternative food systems are minimizing the distance between food production and consumption what allows us to protect the environment.
Detailed comments on the article:
- Keywords do not match the article title.
- Introduction introduces well to the subject.
- Methodology has been carefully described although the statistical interpretation of results should be encouraged.
- Results need improvement – (1) tables and figures require statistical interpretation in the area of content, where no statistical interpretation is given, (2) you should discuss how to interpret the differences for the two areas and why they occur if the areas were studied as similar, (3) why total consumption is not given in table 3, (4) state in the content of the table 4 what means e.g. 1/2 cup.
- Discussion was based on 4 articles, there are repetitions to the results, which can be improved.
Round 2
Reviewer 2 Report
Thank the Authors for completing and improving the content according to the comments in review 1. The Authors changed the keywords, corrected the presentation of results, supplemented the content of the chapters. Currently, I recommend work for approval by the Editors.